# MOTION-INDUCTIVE SELF-SUPERVISED OBJECT DISCOVERY IN VIDEOS

## ABSTRACT

In this paper, we consider the task of unsupervised object discovery in videos. Previous works have shown promising results via processing optical flows to segment objects. However, taking flow as input brings about two drawbacks. First, flow cannot capture sufficient cues when objects remain static or partially occluded. Second, it is challenging to establish temporal coherency from flow-only input, due to the missing texture information. To tackle these limitations, we propose a model for directly processing consecutive RGB frames, and infer the optical flow between any pair of frames using a layered representation, with the opacity channels being treated as the segmentation. Additionally, to enforce object permanence, we apply temporal consistency loss on the inferred masks from randomly-paired frames, which refer to the motions at different paces, and encourage the model to segment the objects even if they may not move at the current time point. Experimentally, we demonstrate superior performance over previous state-of-the-art methods on three public video segmentation datasets (DAVIS2016, SegTrackv2, and FBMS-59), while being computationally efficient by avoiding the overhead of computing optical flow as input.

## 1 INTRODUCTION

Representing the visual scene with objects as the basic elements has long been considered a fundamental cognitive ability of the intelligent agent, for it enables understanding and interaction with the world more efficiently, for example, combinatorial generalization in novel settings (Tenenbaum et al., 2011). Although it remains somewhat obscure at the level of neurophysiology on exactly how humans discover the objects in a visual scene in the first place, it is a consensus that motion seems to play an indispensable role in defining and discovering the objects from the scene. For example, in 1923, Wertheimer introduced the common fate principle that elements moving together tends to be perceived as a group (Wertheimer, 1923); while later Gibson claimed the independent motion has even been treated as one attribute to define an object visually (Gibson & Carmichael, 1966). Grounded on the above assumptions, the recent literature has witnessed numerous works with different models proposed for segmenting the moving objects via unsupervised learning (Yang et al., 2019; 2021b;a; Liu et al., 2021).

Exploiting optical flows for object discovery naturally incurs two critical limitations: *First*, objects in videos may stop moving or be partially occluded at any time point, leaving no effective cues for their existence in the flow field; *Second*, computing optical flow from a pair of frames refers to a lossy encoding procedure, that poses a significant challenge for establishing temporal coherence, due to the lack of effective texture information. In contrast, adopting RGB frame sequences poses a few clear advantages. The most obvious one is that, while objects do not necessarily move all the time, the property of temporal coherence in RGB space naturally guarantees a preliminary understanding of object permanence; Additionally, the rich textures in the appearance stream give more distinctive patterns than those in motion, allowing to better identify and distinguish the different objects. Last but not least, processing RGB streams still enables a faster processing speed than using optical flow.

In this paper, our goal is to train a video segmentation model that can discover the moving objects within a sequence of RGB frames, in the form of segmentation. In specific, our proposed model first encodes consecutive frames independently, into a set of frame-wise visual features, that is followed by a temporal fusion with a Transformer encoder. To localise the moving objects, we randomly

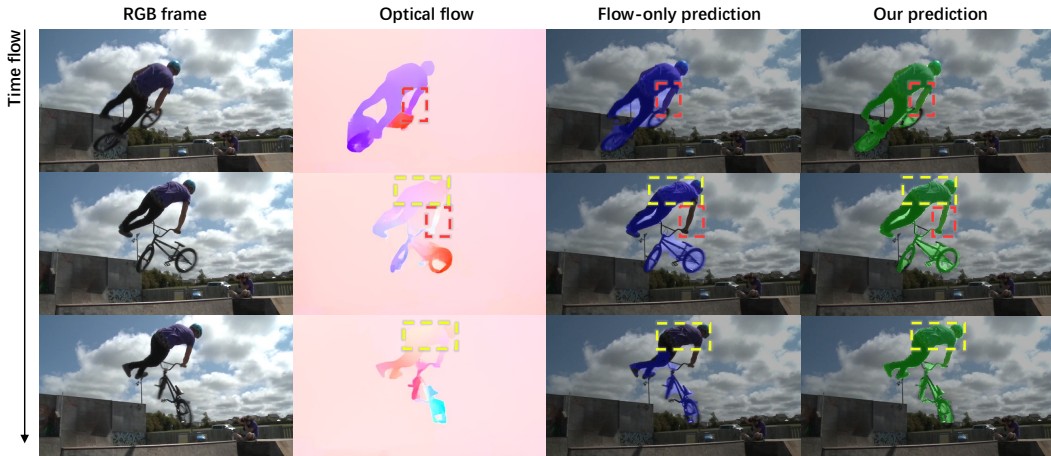

Figure 1: Illustration about Bicycle Motocross (BMX) sequence on SegTrackv2 (Li et al., 2013). The red boxes and the yellow boxes refer to the arm and the back of the player, respectively. Flow-only method (Yang et al., 2021a) fails to track the same region in a temporal consistent fashion since it derives the foreground region directly from current optical flow. However, our methodology of processing a *RGB* video clip develops a sense of object permanence and solves the issue.

pair the visual features from two frames and pass them into a frame comparator module, effectively establishing the relative motion between frames. Inspired by Yang et al. (2021a), we decode the motion features into optical flows with a dual-layered representation, with the opacity weight of each layer treated as the segmentation mask. At training time, we exploit an off-the-shelf optical flow estimator, *e.g.*, RAFT (Teed & Deng, 2020), as the induction for flow reconstruction. To develop the property of object permanence, we enforce a temporal consistency on the inferred segmentation masks, which encourages the model to mine effective texture information from the RGB sequence and keep track of the objects even if they may be static at the current time point.

In short, we summarize the contributions in this paper: *First*, we introduce the **M**otion-inductive **O**bject **D**iscovery (**MOD**) model, a simple architecture for discovering the moving objects in videos, by directly processing a set of consecutive RGB frames. *Second*, we propose a self-supervised proxy task that is used to train the architecture without relying upon any manual annotation. To overcome the challenge from flow-based methods, *i.e.*, objects may stay static or move slowly, we adopt a random-paired policy and restrain the temporal consistency. *Third*, we conduct a series of ablation studies to validate each key component of our method, such as the temporal consistency of random-paired flow. While evaluating three public benchmarks, we demonstrate superior performance over existing approaches on DAVIS2016 (Perazzi et al., 2016), SegTrackv2 (Li et al., 2013), and FBMS-59 (Ochs et al., 2013), with considerable speed-up during the inference procedure.

## 2 RELATED WORK

**Video Object Segmentation.**   How to segment objects coherently in one video sequence has extended the topic of instance segmentation in the image. There is a great amount of work about video object segmentation (VOS) in recent decades (Caelles et al., 2017; Hu et al., 2017; Fan et al., 2019; Dutt Jain et al., 2017; Lai & Xie, 2019; Maninis et al., 2018; Oh et al., 2019; Voigtlaender et al., 2019; Caelles et al., 2017; Perazzi et al., 2017; Hu et al., 2018; Li & Loy, 2018; Bao et al., 2018; Voigtlaender et al., 2019; Johnander et al., 2019). Recently, the research on getting rid of the dense annotation and designing more effective self-supervised algorithms has attracted more and more interest in the computer vision community including VOS (Xu & Wang, 2021; Jabri et al., 2020; Lai et al., 2020; Li et al., 2019; Vondrick et al., 2018; Lu et al., 2020; Wang et al., 2019; Kipf et al., 2022). For VOS, there are two mainstream protocols to evaluate the learned model. One is semi-supervised video object segmentation, the other is unsupervised video object segmentation. Given the first-frame mask of the objects of interest, semi-supervised VOS tracks those objects in subsequent frames, while unsupervised VOS directly segments the most salient objects from the background without any reference. These two protocols are defined in the inference phase, meaning

methods could leverage ground truth annotations in the training stage. In this paper, we don't use any kinds of manual annotations for either training or evaluation.

**Motion Segmentation.** As the name suggests, the aim of motion segmentation is to discover moving objects. One line of the work (Brox & Malik, 2010; Fragkiadaki et al., 2012; Ochs & Brox, 2012; Ochs et al., 2013; Lezama et al., 2011; Keuper et al., 2015) formulates motion as the point trajectory to take advantage of long-range temporal information so that segmentation results can be acquired by grouping the trajectories. Later, deep learning methods take over the area (Tokmakov et al., 2017b;a; Xie et al., 2019; Yang et al., 2019; 2021a; Choudhury et al., 2022; Yang et al., 2021b; Ye et al., 2022a; Wang et al., 2022). Tokmakov et al. (2017b) adopts a two-stream network that ingests both RGB and optical flow. Then they realize a memory mechanism by the convolutional recurrent unit to enhance the visual cues. CIS (Yang et al., 2019) achieves fully unsupervised motion segmentation which discards the supervision of annotated masks during training. By formulating a min-max game of mutual information, the generator is asked to create foreground segments that are as unrelated as possible to the background. AMD (Liu et al., 2021) minimizes the warp synthesis error to train appearance and motion pathways without any supervision. The most similar work to ours is MG (Yang et al., 2021a), which solely leverages the optical flow to separate the pixels via cross attention mechanism (Locatello et al., 2020). Compared to MG (Yang et al., 2021a), we keep reserved on the module design and the training recipe to demonstrate the improvement brought by our method is purely from taking consecutive RGB frames instead of optical flow. Recent work GWM (Choudhury et al., 2022) also utilizes RGB images and adopt the supervision from optical flow but their model ingests a single image to segment the foreground and fails to consider temporal coherency.

**Object Discovery.** There is rich literature on identifying salient objects without explicit supervision, known as object discovery. There exist a series of works that aim to learn object-centric representations in images (Locatello et al., 2020; Lin et al., 2020; Jiang & Ahn, 2020; Greff et al., 2019; Emami et al., 2021; Burgess et al., 2019; Crawford & Pineau, 2019; Engelcke et al., 2019; 2021). Typically, IODINE (Greff et al., 2019) develops iterative variational inference to separate different objects. Locatello et al. (2020) proposes slot attention to iteratively update latent object representations. Further, a line of works (Zablotskaia et al., 2020; Min et al., 2021; Kosiorek et al., 2018; Kipf et al., 2022; Jiang et al., 2019; Kabra et al., 2021; Crawford & Pineau, 2020; Besbinar & Frossard, 2021; Bear et al., 2020; Bao et al., 2022; Ye et al., 2022b; Yang et al., 2021a) extend object-centric learning to video domain. Most of these approaches incorporate motion cues into the reconstruction task and perform well in moving object segmentation but perform poorly in processing static objects. While in this work, we adopt slot attention to form optical flow reconstruction bottleneck and perceive both dynamic and static instances through RGB clip input in a temporally consistent way.

## 3 MOTION-INDUCTIVE OBJECT DISCOVERY (MOD) MODEL

In this section, we detail our MOD model, which processes a set of consecutive RGB frames and automatically discovers the moving objects in the form of segmentation. An overview of the training procedure can be seen in Figure 2, where the visual features computed from individual frames are temporally fused, and randomly paired together, for decoding the optical flows between two corresponding frames. Taking inspiration from the motion grouping (Yang et al., 2021a), we also adopt a dual-layered representation for the output flow, with the foreground and background flows being reconstructed separately, and later composited with the inferred opacity masks (soft segmentation). In the following section, we will detail each key component in our proposed architecture.

### 3.1 SPATIAL-TEMPORAL VISUAL ENCODER

To start with, our model takes a short video clip as input, *i.e.*, $v = \{x_1, \cdots, x_T\}, v \in \mathbb{R}^{T \times H \times W \times 3}$, consisting of a set of RGB frames, the frame-wise visual representations are computed with a shared visual encoder $\Phi_{\text{enc}}$. Formally, the output feature map $o_t$ at timestamp $t$ ($1 \leq t \leq T$) can be obtained:

$$o_t = \Phi_{\text{enc}}(x_t) \in \mathbb{R}^{h \times w \times d}, \tag{1}$$

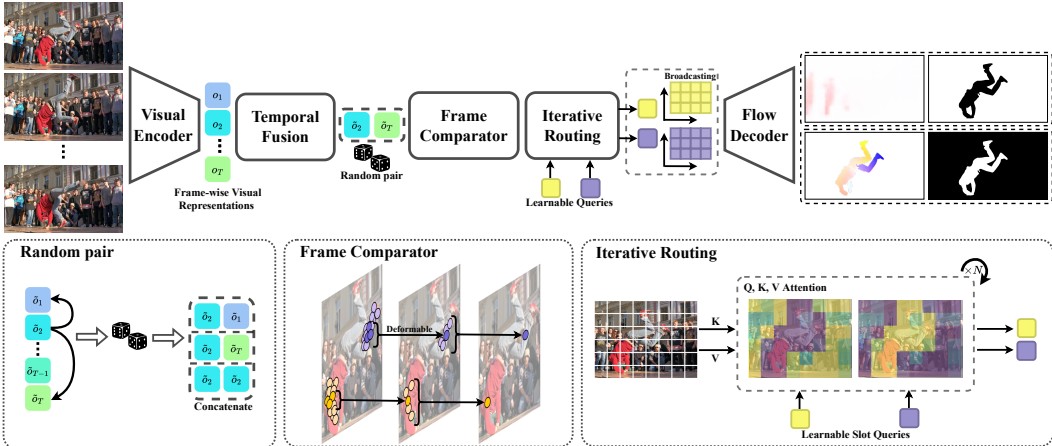

Figure 2: **Model architecture**. Our model first extracts spatial features of consecutive frames by the visual encoder. To jointly model the temporal relation, we aggregate and interact among the multi-frame features with late-fusion. We randomly pair two frames' visual representations and pass them into the frame comparator to encode relative motion. Then, we decode the flow from random-paired frames. Through iterative routing, we adopt a dual-layer representation for the flow reconstruction, *i.e.*, outputting the foreground and background flows separately, and composing them with inferred opacity weights. The whole training procedure does not require supervision from any mask annotations.

where $h, w, d$ denotes the dimension of the height, width, and channel, respectively. Till this point, to build the temporal dependency between multiple visual frames, a global fusion module is introduced, via a standard Transformer Encoder layer:

$$\{\tilde{o}_1, \ldots, \tilde{o}_T\} = \Phi_{\text{temp}}(\{o_1 + \texttt{pe}_1, \cdots, o_T + \texttt{pe}_T\}), \tag{2}$$

where $\texttt{pe}$ refers to the learnable spatial-temporal positional encodings, and the output $\tilde{o}_t \in \mathbb{R}^{h \times w \times d}$ remains the same dimension as input.

By taking the multiple RGB frames as input, our proposed visual encoder can explicitly consider the temporal coherence within the video clip. **Note that**, such seemingly simple design poses two critical differences from previous work on motion-driven object discovery (Yang et al., 2019; 2021a), where only frame-wise optical flow is adopted: *First*, using RGB frames as input can drastically reduce the computation latency at inference time. The throughput of our model without computing dense optical flow reaches round **100 fps** on a standard 32GB Tesla V100 GPU while prior works need to calculate flow at first. RGB provides more semantic information than flows for the model to exploit, *i.e.*, including not only the object's shape but also its texture. *Second*, processing multiple frames contributes to the development of a sense of object permanence within the video clip, *i.e.*, the understanding that items or people still exist even when they cannot be perceived explicitly. Therefore, even though the objects in videos may stop moving or be partially occluded at any time point, they can still be effectively segmented with the temporal cues.

### 3.2 RANDOM-PAIRED FRAME COMPARATOR

Till this point, we consider building the relative motion between any two visual frames within the video clip, from reference frame $i$ to target frame $j$, *i.e.*, $f_{i \rightarrow j}$. In specific, we select the visual representation of the two corresponding frames, *i.e.*, $\tilde{o}_i$ and $\tilde{o}_j$, concatenate them along the feature dimension, and feed it into a comparator module:

$$f_{i \rightarrow j} = \Phi_{\text{comp}}(\text{concat}(\tilde{o}_i, \tilde{o}_j)) \in \mathbb{R}^{h \times w \times d}, \tag{3}$$

where $\Phi_{\text{comp}} : \mathbb{R}^{2d} \rightarrow \mathbb{R}^d$ consists of multiple deformable convolutional layers (Zhu et al., 2019) followed by a series of Transformer Encoders, that dynamically construct the feature representation for later estimating relative motion between frames while reducing the feature dimensions at the same time.

**Discussion.** With such a design of random-paired frame comparator, our MOD is capable of modeling the relative motion between any two randomly sampled frames out of the video clip ($T$ frames), accounting for a total of $T^2$ frame pairs for forward, backward, single, or multi-step motions. Note that, we do not distinguish the case of $i \neq j$ and $i = j$. Specifically, the former encourages the model to discover the relative motion between two different frames, while the latter refers to an extreme case that neither objects nor camera is moving, and no motion cues are available, thus enforcing the model to discover objects via temporal coherence, *i.e.*, objects that moves in any frame along the video should also be discovered in static frames.

### 3.3 Dual-layered Flow Decoder

To decode the features into the form of optical flow, we adopt several slot attention layers (Locatello et al., 2020) with two learnable queries, *i.e.*, termed as slot vectors, iteratively attending the output visual features from the comparator, and decoded into the optical flow between any two frames with a dual-layered representation. In detail, a slot attention module acts similarly to a Transformer Decoder, with the only exception being that the normalisation is computed along the slot side, thus each slot competes to take over the pixels. In each iteration, given two slot vectors as $S \in \mathbb{R}^{2 \times d}$ and visual feature maps $\tilde{o}_t \in \mathbb{R}^{hw \times d}$, we use three linear projections to compute the `query`, `key` and `value`, *i.e.*, $Q \in \mathbb{R}^{2 \times d}, K, V \in \mathbb{R}^{hw \times d}$. Thereafter, we can obtain the weights matrix $W \in \mathbb{R}^{2 \times hw}$ and normalise along the slot dimension, *i.e.*,

$$\widetilde{W}_{s,\cdot} = \exp(W_{s,\cdot}) / \sum_l \exp(W_{l,\cdot}), \text{ where } W = \frac{1}{\sqrt{d}} QK^T. \tag{4}$$

Then, we gain the next iteration's slot vectors by aggregating the values $V$ and passing them into a Gated Recurrent Unit (GRU) (Cho et al., 2014), *i.e.*,

$$S := \text{GRU}(\text{inputs} = AV, \text{states} = S), \text{ where } A_{\cdot,s} = \frac{\widetilde{W}_{\cdot,s}}{\sum_l \widetilde{W}_{\cdot,l}} \in \mathbb{R}^{2 \times (h \times w)}. \tag{5}$$

We iterate the whole routing process for $N$ times. In this way, the entities of similar RGB and flow patterns are grouped together and distinct pixels are separated by two slots. Eventually, we broadcast the final outputted slots into $G = \{G^s \in \mathbb{R}^{h \times w \times d}\}_{s=1}^2$ added with learnable spatial positional embeddings to construct the optical flow. A flow decoder $\Phi_{\text{dec}}$ consisting Transformer encoders and up-sampling layers takes the slot grids as input, and outputs dual layers of optical flows $\{\widetilde{I}^s \in \mathbb{R}^{H \times W \times 3}\}_{s=1}^2$ [1] and their opacity weights $\{\alpha^s \in \mathbb{R}^{H \times W \times 1}\}_{s=1}^2$:

$$\{\widetilde{I}^s, \alpha^s\}_{s=1}^2 = \Phi_{\text{dec}}(G), \tag{6}$$

where $\alpha^s \in [0, 1]^{H \times W \times 1}$ is normalized across two slots via $\text{softmax}$ function. Noted that thanks to the softmax function, the alpha value for foreground could be close to zero when the foreground object is missing in part of the video. For a given frame pair $(i, j)$, their relative flow $\widetilde{I}_{i \to j}$ can be computed via:

$$\widetilde{I}_{i \to j} = \sum_{s=1}^2 \alpha_{i \to j}^s \otimes \widetilde{I}_{i \to j}^s, \tag{7}$$

where $\otimes$ denotes the element-wise multiplication. At inference time, we adopt the binarized opacity weights $\alpha^s$ as the object segmentation masks.

### 3.4 Training

In this section, we describe the training procedure for the proposed model, on the raw videos without using manual annotations for the object segmentations. In general, the training loss is composed of three components, namely, flow reconstruction, temporal consistency, and entropy minimisation.

---

[1]Note that following the common practise, we also output 3-channel RGB images, that refers to a transformation from the traditional 2-channel optical flow based on the color wheel proposed in Sun et al. (2018).

**Flow Reconstruction.** As the main objective for optimisation, we use the flow reconstruction, where we adopt an off-the-shelf optical flow estimator, for example, RAFT (Teed & Deng, 2020), to estimate the flow between any two frames in the video. We minimise the discrepancy between the dual-layer flow reconstruction and the output from the existing flow estimator:

$$\mathcal{L}_{\text{recon}}^{i\to j} = \frac{1}{|\Omega|} \sum_{u\in\Omega} |I_{i\to j}(u) - \widetilde{I}_{i\to j}(u)|_2, \tag{8}$$

where $\Omega = \{1, \cdots, H\} \times \{1, \cdots, W\}$ represents the spatial lattice, and a $|\cdot|_2$ denotes the L2 norm. In practice, the reconstruction constraint for zero flow when $i = j$ is dropped due to the distribution gap between dynamic flow and static flow, resulting in the divergence of the training. **Note that**, flow estimator is only used for model training, at inference time, our proposed architecture directly processes the RGB video clips.

**Temporal Consistency.** In order to build up temporal consistency within the input video, the pair of motion embeddings $f_{i\to j}, f_{i\to k}$ that starts from the same reference frame $i$, are passed through the flow decoder to reconstruct the optical flow between two corresponding frames. Note that, as $1 \le j, k \le T$ are randomly sampled at every training iteration, the output flow will refer to the motion at a different pace. However, the predicted alpha weights for flow composition denote the soft segmentation for the same objects in the $i$-th frame, thus remaining consistent. In specific, the two inferred masks $\{\alpha_{i\to j}^s\}_{s=1}^2, \{\alpha_{i\to k}^s\}_{s=1}^2$ are enforced to pull closer by minimising mean-squared error $\mathcal{L}_{\text{cons}}$, *i.e.*,

$$\mathcal{L}_{\text{cons}} = \frac{1}{T} \sum_{i=1}^T \mathcal{L}_{\text{cons}}^i \quad \text{where } \mathcal{L}_{\text{cons}}^i = \frac{1}{2|\Omega|} \sum_{u\in\Omega} \sum_{s=1}^2 |\alpha_{i\to j}^s(u) - \alpha_{i\to k}^s(u)|^2. \tag{9}$$

**Entropy Minimisation.** Lastly, we impose a pixel-wise entropy regularisation on inferred masks, that is zero if the alpha channels are one-hot, and maximum when they are of equal probability. Intuitively, this helps encourage the masks to be binary, which aligns with our goal in obtaining segmentation masks:

$$\mathcal{L}_{\text{entro}}^{i\to j} = \frac{1}{2|\Omega|} \sum_{u\in\Omega} \sum_{s=1}^2 -\alpha_{i\to j}^s(u) \log(\alpha_{i\to j}^s(u)). \tag{10}$$

**Total Loss.** Accordingly, we rewrite aforementioned reconstruction loss $\mathcal{L}_{\text{recon}}$ and entropy regulatization $\mathcal{L}_{\text{entro}}$ in summed version:

$$\mathcal{L}_{\text{recon}} = \frac{1}{2T} \sum_{i=1}^T \mathcal{L}_{\text{recon}}^{i\to j_i} + \mathcal{L}_{\text{recon}}^{i\to k_i}; \tag{11}$$

$$\mathcal{L}_{\text{entro}} = \frac{1}{2T} \sum_{i=1}^T \mathcal{L}_{\text{entro}}^{i\to j_i} + \mathcal{L}_{\text{entro}}^{i\to k_i}. \tag{12}$$

The total loss for training our model can thus be computed as:

$$\mathcal{L}_{\text{tot}} = \lambda_{\text{r}} \mathcal{L}_{\text{recon}} + \lambda_{\text{e}} \mathcal{L}_{\text{entro}} + \lambda_{\text{c}} \mathcal{L}_{\text{cons}}, \tag{13}$$

where we set $\lambda_{\text{r}} = 100, \lambda_{\text{e}} = \lambda_{\text{c}} = 0.01$ at the beginning of the training. We notice the model to be fairly robust to these hyper-parameters.

## 4 EXPERIMENTAL SETUP

In the experiments setup, we first introduce the benchmarks and then elaborate on implementation details.

### 4.1 DATASETS

We benchmark on three popular datasets designed for video object segmentation. **DAVIS2016** (Perazzi et al., 2016) consists of 50 high quality videos, 3455 frames in total. Every frame is annotated

with a pixel-accurate segmentation mask. **SegTrackv2** (Li et al., 2013) contains 14 sequences and 947 fully-annotated frames. Each sequence involves 1-6 moving objects and presents challenges including motion blur, appearance change, complex deformation, occlusion, slow motion, and interacting objects. **FBMS-59** (Ochs et al., 2013) has 59 sequences with greatly varied resolution and annotates every 20th frame. Many sequences contain multiple moving objects. Following previous evaluation metric (Yang et al., 2019; Xie et al., 2022), we merge objects of SegTrackv2 and FBMS-59 into one single object for video object segmentation. We evaluate the pixel-wise segmentation through Jaccard index $\mathcal{J}$, also called Intersection over Union (IoU). Following prior arts (Yang et al., 2019; 2021a), we compute the mean per frame over the test set and merge multi-object annotation into single unified segmentation.

## 4.2 IMPLEMENTATION DETAILS

For data input, we sample $T = 7$ consecutive frames as the input clip. Each frame is resized to $192 \times 384$ and the estimated optical flow is computed by RAFT (Teed & Deng, 2020), which is pre-trained on the synthetic dataset (Mayer et al., 2016).

To compute spatial-temporal visual representation, we adopt the first three stages of a SwinV2-T as the frame encoder $\Phi_{\text{enc}}$, which is then followed by a standard Transformer Encoders with 8 heads (Vaswani et al., 2017) as temporal fusion module $\Phi_{\text{temp}}$. For the frame comparator $\Phi_{\text{comp}}$, we use two deformable convolutional layers in the company with three standard Transformer Encoders with 8 heads to process the pixel transformation. Then, we choose $N = 5$ iteration in total for the iterative routing in slot attention. Lastly, we utilize three stages of SwinV2 blocks with the linear patch expanding layers as the flow decoder $\Phi_{\text{dec}}$ (Cao et al., 2021).

As for training, we adopt AdamW optimizer (Loshchilov & Hutter, 2018) with learning rate $4 \times 10^{-5}$. The model is trained from scratch without any pretrained weights, for a total of 300k iterations. At inference time, we adopt the overlapping temporal sliding window to ensemble the segmentation masks. We average the resultant masks obtained by the whole temporal segments. We propose two protocols to evaluate our results. Besides measuring the masks without any post-processing, we also apply **test-time adaptation** with the help of the self-supervised DINO-pretrained ViT (Caron et al., 2021). Without any fine-tuning, the pretrained ViT can propagate the masks as noisy annotations to the whole frames in the same manner as CRW (Jabri et al., 2020). We refine the masks further with CRF (Lafferty et al., 2001). For more detailed technical information, please refer to the supplementary materials.

## 5 RESULTS

In this section, we compare primarily with several top-performing approaches trained without human annotations, for example, OCLR (Xie et al., 2022), MG (Yang et al., 2021a), CIS (Yang et al., 2019), *etc*. However, as the architecture, modality, input resolution, and post-processing protocols are all different, we try our best to conduct the comparison as fairly as possible.

## 5.1 ABLATION STUDY

We conduct all ablation studies on DAVIS2016 and vary one variable each time, as shown in Table 1.

**Temporal Fusion $\Phi_{\text{temp}}$ and Frame Comparator $\Phi_{\text{comp}}$.** As shown by Ours-A and Our-C, the performance degrades significantly without temporal fusion, demonstrating the importance of building up global temporal dependency. Also, indicated by Ours-B, we find the model fails to converge when removing the component of frame comparator $\Phi_{\text{comp}}$. It meets expectations because frame comparator $\Phi_{\text{comp}}$ is the sole module in charge of relative motion estimation. Without it, the model cannot reconstruct optical flow thus leading to divergence.

**Number of Frames $T$.** While comparing Ours-A, Ours-D, and Ours-E, there is a clear trend that increasing the frame number boosts the segmentation quality, which coincides with our intuition that incorporating a wider temporal receptive field can enhance the sense of temporal coherence and object permanence. Due to the limited computational memory, we only set $T = 7$ as the maximum frame number in the paper. A promising performance is expected when inputting more frames.

| Model | $\Phi_{\text{temp}}$ | $\Phi_{\text{comp}}$ | T | $\mathcal{L}_{\text{cons}}$ | $\mathcal{L}_{\text{entro}}$ | DAVIS($\mathcal{J} \uparrow$) |
|---|---|---|---|---|---|---|
| Ours-A | ✓ | ✓ | 7 | ✓ | ✓ | **73.9** |
| Ours-B | - | ✗ | 7 | ✓ | ✓ | fail |
| Ours-C | ✗ | ✓ | 7 | ✓ | ✓ | 68.3 |
| Ours-D | ✓ | ✓ | 3 | ✓ | ✓ | 66.4 |
| Ours-E | ✓ | ✓ | 5 | ✓ | ✓ | 68.2 |
| Ours-F | ✓ | ✓ | 7 | ✗ | ✗ | 60.4 |
| Ours-G | ✓ | ✓ | 7 | ✗ | ✓ | 65.6 |
| Ours-H | ✓ | ✓ | 7 | ✓ | ✗ | 69.5 |

Table 1: Ablation studies on temporal fusion ($\Phi_{\text{temp}}$), frame comparator ($\Phi_{\text{comp}}$), the number of input frames ($T$), temporal consistency ($\mathcal{L}_{\text{cons}}$), and entropy loss ($\mathcal{L}_{\text{entro}}$).

| | Training | Inference | | | $\mathcal{J}$ (Mean) ↑ | | |
|---|---|---|---|---|---|---|---|
| Model | Sup. | RGB | Flow | p.p. | DAVIS2016 | SegTrackv2 | FBMS-59 |
| SAGE (Wang et al., 2017) | None | ✓ | ✓ | ✗ | 42.6 | 57.6 | 61.2 |
| NLC (Faktor & Irani, 2014) | None | ✓ | ✓ | ✓ | 55.1 | 67.2 | 51.5 |
| CIS (Yang et al., 2019) | None | ✓ | ✓ | ✓ | 71.5 | 62.5 | 63.5 |
| AMD (Liu et al., 2021) | None | ✓ | ✗ | ✗ | 57.8 | 57.0 | 47.5 |
| SIMO (Lamdouar et al., 2021) | Syn. | ✗ | ✓ | ✓ | 67.8 | 62.0 | - |
| MG (Yang et al., 2021a) | None | ✗ | ✓ | ✗ | 68.3 | 58.6 | 53.1 |
| OCLR (Xie et al., 2022) | Syn. | ✗ | ✓ | ✗ | 72.1 | 67.6 | 65.4 |
| **MOD (w/o post-processing)** | None | ✓ | ✗ | ✗ | 73.9 | 62.2 | 61.3 |
| **MOD (test-time adaptation)** | None | ✓ | ✗ | ✓ | 79.2 | 69.4 | 66.9 |
| FSEG (Dutt Jain et al., 2017) | GT | ✓ | ✓ | - | 70.7 | 61.4 | 68.4 |
| COSNet (Lu et al., 2019) | GT | ✓ | ✗ | - | 80.5 | 49.7 | 75.6 |
| MATNet (Zhou et al., 2020) | GT | ✓ | ✓ | - | 82.4 | 50.4 | 76.1 |
| $D^2$Conv3d (Schmidt et al., 2022) | GT | ✓ | ✗ | - | 85.5 | - | - |

Table 2: Quantitative comparison on unsupervised video object segmentation. We compare our method on three standard datasets, DAVIS2016, SegTrackv2, and FBMS-59. Sup. refers to the supervision, including None, Synthetic (Syn.), and Ground Truth (GT). p.p. is short for **post-processing** (e.g., CRF (Lafferty et al., 2001)).

**Temporal Consistency $\mathcal{L}_{\text{cons}}$ and Entropy Regularisation $\mathcal{L}_{\text{entro}}$.** Lastly, comparing Ours-G and Ours-A, we observe that the performance increases considerably with temporal consistency. It manifests the validity of our design motivation, temporal consistency conduces to persistently tracking the object. The entropy regularisation is also indispensable shown by Ours-H and Ours-A.

## 5.2 COMPARISON WITH STATE-OF-THE-ART

We show the comparison with state-of-the-art in Table 2. On DAVIS2016, MOD achieves 73.9% mIOU without any post-processing, exceeding MG (Yang et al., 2021a) by a large margin (+5.8%). Compared to the latest method OCLR (Xie et al., 2022) which fabricates a synthesized dataset to train its model, our method still surpasses it only using the information of the DAVIS dataset itself. On another two benchmarks SegTrackv2 and FBMS-59, MOD also beats MG (Yang et al., 2021a), which only leverages single-step flows to decompose foreground and background, by +3.6% and +8.2%, respectively. The superior experimental results demonstrate that our methodology of multi-frame reasoning benefits moving object discovery. Furthermore, equipped with DINO-pretrained ViT, a further performance gain is observed on all three benchmarks, which is even more competitive with current supervised approaches.

## 5.3 QUALITATIVE RESULTS

In Figure 3, we present several qualitative illustrations of the model. It can be seen that our results are robust to the noticeable background flow signal (drift-chicane sequence in the second column) and

estimate more accurate boundaries when single-step foreground flow cannot represent exact object shape (breakdance and dance-twirl sequences in middle) compared to MG (Yang et al., 2021a). It demonstrates inferring the masks by associating a bunch of RGB features well resolves the limitation of the usage of flow. Moreover, in virtue of temporally consistent cues, our model handles occlusion well shown in the libby sequence at the rightmost column, which could be hard for the flow-only method to maintain the object shape constantly.

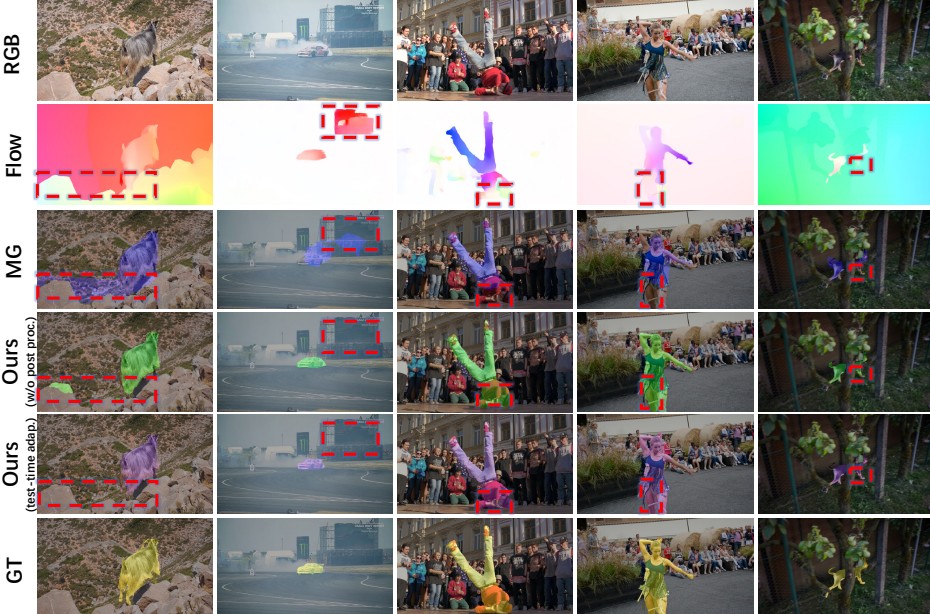

Figure 3: Qualitative results of object video segmentation on DAVIS2016. MG refers to Yang et al. (2021a). Red boxes outline the corresponding difference.

## 5.4 LIMITATIONS

Though we demonstrate that associating multiple frames stimulates the comprehensive sense of objectness, which is proved by the superior experimental performance across the prior arts, there still exist limitations and room for improvement. *First*, our method uses a number of consecutive frames, which challenges computational memory. How to utilize pretrained features for reducing training expenses would be meaningful. *Second*, how to segment multiple objects remains unresolved. We display a preliminary result in supplementary material. It will be promising to exploit the semantic information from RGB to discriminate different foreground objects. We leave them as the feature work. Despite these limitations, the approach has convincingly manifested the value of considering textural information and processing RGB frames as a whole.

## 6 CONCLUSION

In this paper, we propose a self-supervised model for video object discovery. The model takes a set of consecutive RGB frames as input and generates the segmentation mask for the moving objects in the video. At training time, the model is tasked to reconstruct the optical flow between any pair of frames, through a layered representation with the opacity channels being treated as the segmentation. To encourage the model to capture the objects even when they may be static at a certain time point, a temporal consistency loss is enforced on the inferred masks on the randomly-paired frames. As a consequence, we demonstrate superior performance over previous state-of-the-art methods on three public video segmentation datasets (DAVIS2016, SegTrackv2, and FBMS-59), while being computationally efficient by avoiding the overhead of computing optical flow.

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

## A   MORE IMPLEMENTATION DETAILS

In this section, we list the architecture details and training settings. Codes and models will be released publicly.

### A.1   VISUAL ENCODER

The visual encoder contains the first three stages of Swin-Tiny V2. We tabulate the workflow in Table 3.

| stage | operation | output sizes |
|---|---|---|
| input | - | $3 \times 192 \times 384$ |
| PatchEmbed | $4 \times 4$, stride 4, 96 | $96 \times 48 \times 96$ |
| SwinBlock1 | $\begin{bmatrix} h = 3 \\ ws = 12 \end{bmatrix} \times 2$ | $96 \times 48 \times 96$ |
| DownSample1 | 192 | $192 \times 24 \times 48$ |
| SwinBlock2 | $\begin{bmatrix} h = 6 \\ ws = 12 \end{bmatrix} \times 2$ | $192 \times 24 \times 48$ |
| DownSample2 | 384 | $384 \times 12 \times 24$ |
| SwinBlock3 | $\begin{bmatrix} h = 12 \\ ws = 12 \end{bmatrix} \times 6$ | $384 \times 12 \times 24$ |

Table 3: Architecture of visual encoder. $h$ stands for the number of attention heads while $ws$ refers to window size.

### A.2   FRAME COMPARATOR

The frame comparator possesses two deformable convolutional layers, with ReLU operation in between and three Transformer Encoder blocks. We tabulate the workflow in Table 4.

| stage | operation | output sizes |
|---|---|---|
| input | - | $768 \times 12 \times 24$ |
| DeformConv1 | $3 \times 3$, 768 | $768 \times 12 \times 24$ |
| ReLU | - | $768 \times 12 \times 24$ |
| DeformConv2 | $3 \times 3$, 384 | $384 \times 12 \times 24$ |
| Transformer | $\begin{bmatrix} h = 8 \\ 384 \end{bmatrix} \times 3$ | $384 \times 12 \times 24$ |

Table 4: Architecture of frame comparator. $h$ represents the number of attention heads

### A.3   FLOW DECODER

The frame comparator consists of three stages of SwinV2 block added with linear expanding layers. We tabulate the workflow in Table 5.

### A.4   TRAINING DETAILS

For all datasets, we train with a batch size of 2. To train more efficiently, we sample three flow pairs $(i \rightarrow j)$ for a given frame $i$; one is static replication $i = j$, another two are motion pair $i \neq j$. We apply temporal consistency first on the masks from two motion pairs, then pull the static mask to the average of two dynamic masks closer. We linearly warm up the learning rate for the first 1k iterations. Besides, for every 100k iterations, we decay the learning rate by half and increase the scale of temporal consistency $\lambda_c$ and entropy regularisation $\lambda_e$ by the factor of 5. In the default setting, we train for about 3 days on 8 standard Tesla V100 GPUs with 32GB memory each.

| stage | operation | output sizes |
|---|---|---|
| input | - | $384 \times 12 \times 24$ |
| SwinBlock1 | $\begin{bmatrix} h = 12 \\ ws = 12 \end{bmatrix} \times 2$ | $384 \times 12 \times 24$ |
| PatchExpand1 | 768 | $192 \times 24 \times 48$ |
| SwinBlock2 | $\begin{bmatrix} h = 6 \\ ws = 12 \end{bmatrix} \times 2$ | $192 \times 24 \times 48$ |
| PatchExpand2 | 384 | $96 \times 48 \times 96$ |
| SwinBlock3 | $\begin{bmatrix} h = 3 \\ ws = 12 \end{bmatrix} \times 2$ | $96 \times 48 \times 96$ |
| PatchExpand3 | 1536 | $96 \times 192 \times 384$ |
| outConv | $5 \times 5$, stride 1, 4 | $4 \times 192 \times 384$ |

Table 5: Architecture of flow decoder. $h$ stands for the number of attention heads while $ws$ refers to window size.

### A.5 TEST-TIME ADAPTATION

Inspired by OCLR (Xie et al., 2022), we adopt test-time adaptation based on RGB sequence to enhance appearance consistency. In detail, we follow existing works on self-supervised tracking (Lai et al., 2020; Jabri et al., 2020) to propagate object masks across time span. The whole process consists of three steps. First, we extract RGB features of each frame with a DINO-pretrained ViT encoder. Then, we select key frames for object mask propagation. Finally, we calculate the affinity matrix between frames and perform mask propagation.

**DINO Feature Extraction.** Given a video sequence $v = \{x_1, ..., x_T\}, v \in \mathbb{R}^{T \times H \times W \times 3}$, we use DINO pretrained ViT-Small encoder with patch size $8 \times 8$ to extract features:

$$\{f_1, ..., f_T\} = \{\Phi(x_1), ..., \Phi(x_T)\}, \quad f_t \in \mathbb{R}^{h \times w \times 384}, \tag{14}$$

where $h = H//8$ and $w = W//8$. The extracted features will be used in the mask propagation step.

**Key Frame Selection.** Given video $v$, our model predicts object mask of each frame as $m = \{\alpha_1, ..., \alpha_T\}, m \in \mathbb{R}^{T \times H \times W \times 1}$. Since the video frames are continuous along the temporal dimension, it is practical to propagate object masks between neighboring frames. The propagation operation is the same as Jabri et al. (2020), the only difference is that we have no ground-truth mask for reference. Therefore, we need to design a mechanism to select object masks of high confidence. To do this, we measure the temporal coherence of predicted object masks for key frame selection. Specifically, for each timestamp $t \in \{3, ..., T-2\}$, we can calculate four propagated masks as:

$$\hat{\boldsymbol{\alpha}}_t = [\text{Mask-prop}(\alpha_{t-2}), \text{Mask-prop}(\alpha_{t-1}), \text{Mask-prop}(\alpha_{t+1}), \text{Mask-prop}(\alpha_{t+2})], \tag{15}$$

where 'Mask-prop' denotes the propagation operation. Then we calculate the average IoU between the original mask and propagated masks as the confidence score:

$$s_t = \frac{\text{IoU}(\hat{\boldsymbol{\alpha}}_t[0], \alpha_t) + \text{IoU}(\hat{\boldsymbol{\alpha}}_t[1], \alpha_t) + \text{IoU}(\hat{\boldsymbol{\alpha}}_t[2], \alpha_t) + \text{IoU}(\hat{\boldsymbol{\alpha}}_t[3], \alpha_t)}{4}. \tag{16}$$

The calculated $s_t$ measures the coherency between $\alpha_t$ and its neighbors. Our empirical studies show that it serves as a reliable signal for key frame selection.

**Object Mask Propagation.** We select timestamps with Top-$k\%$ confidence score as the key reference frames ($k = 15$ on DAVIS2016, $k = 25$ on SegTrackv2, $k = 10$ on FBMS-59). Then we iteratively propagate the object masks with a neighbor temporal window size $n = 7$. Compared to conventional semi-supervised object segmentation which only relies on the first frame as key frame, we have multiple key frames on different temporal positions to correct the accumulated propagation error. In this way, the propagation enhances object permanence across time and boosts performance.

## B RESULTS BREAKDOWN

We include a specific result breakdown in this section. We show the per-sequence results on DAVIS2016 in Table 6, SegTrackv2 in Table 7 and FBMS-59 in Table 8.

| | $\mathcal{J}$ (Mean) ↑ | |
|---|---|---|
| Sequence | w/o post proc. | test-time adap. |
| dog | 80.7 | 87.4 |
| cows | 87.2 | 88.8 |
| goat | 47.5 | 80.6 |
| camel | 85.6 | 86.1 |
| libby | 72.5 | 77.7 |
| parkour | 72.9 | 87.9 |
| soapbox | 84.6 | 86.5 |
| blackswan | 48.9 | 46.9 |
| bmx-trees | 50.1 | 55.8 |
| kite-surf | 55.9 | 62.6 |
| car-shadow | 87.9 | 86.9 |
| breakdance | 82.6 | 76.0 |
| dance-twirl | 82.5 | 85.4 |
| scooter-black | 80.2 | 80.3 |
| drift-chicane | 78.6 | 82.2 |
| motocross-jump | 68.4 | 88.9 |
| horsejump-high | 78.0 | 84.3 |
| drift-straight | 69.2 | 80.0 |
| car-roundabout | 87.7 | 83.9 |
| paragliding-launch | 62.1 | 62.8 |
| frame avg. | 73.9 | 79.2 |

Table 6: Sequence-wise results on DAVIS2016.

| | $\mathcal{J}$ (Mean) ↑ | |
|---|---|---|
| Sequence | w/o post proc. | test-time adap. |
| drift | 41.7 | 40.7 |
| birdfall | 38.2 | 61.5 |
| girl | 76.5 | 82.3 |
| cheetah | 18.4 | 30.1 |
| worm | 52.5 | 74.3 |
| parachute | 90.2 | 92.0 |
| monkeydog | 14.3 | 31.1 |
| hummingbird | 61.2 | 58.8 |
| soldier | 66.3 | 58.8 |
| bmx | 73.7 | 78.8 |
| frog | 80.5 | 76.3 |
| penguin | 63.5 | 62.7 |
| monkey | 46.8 | 77.4 |
| bird of paradise | 85.3 | 85.4 |
| frame avg. | 62.2 | 69.4 |

Table 7: Sequence-wise results on SegTrackv2.

| Sequence | $\mathcal{J}$ (Mean) $\uparrow$ | |
| --- | --- | --- |
| | w/o post proc. | test-time adap. |
| camel01 | 25.8 | 66.9 |
| cars1 | 66.1 | 88.3 |
| cars10 | 31.6 | 33.9 |
| cars4 | 72.9 | 83.2 |
| cars5 | 81.2 | 82.5 |
| cats01 | 80.6 | 79.2 |
| cats03 | 62.0 | 63.4 |
| cats06 | 40.1 | 38.7 |
| dogs01 | 70.6 | 61.2 |
| dogs02 | 62.8 | 82.2 |
| farm01 | 86.7 | 88.9 |
| giraffes01 | 38.6 | 52.2 |
| goats01 | 44.8 | 45.3 |
| horses02 | 64.4 | 77.6 |
| horses04 | 68.5 | 73.5 |
| horses05 | 43.7 | 49.0 |
| lion01 | 60.1 | 71.5 |
| marple12 | 74.7 | 80.3 |
| marple2 | 65.0 | 71.4 |
| marple4 | 79.1 | 91.6 |
| marple6 | 76.0 | 85.1 |
| marple7 | 72.5 | 55.2 |
| marple9 | 87.5 | 97.9 |
| people03 | 76.5 | 48.5 |
| people1 | 72.4 | 80.8 |
| people2 | 80.9 | 83.0 |
| rabbits02 | 50.2 | 58.9 |
| rabbits03 | 39.5 | 55.7 |
| rabbits04 | 47.0 | 53.0 |
| tennis | 63.1 | 71.1 |
| frame avg. | 61.3 | 66.9 |

Table 8: Sequence-wise results on FBMS-59.

| Flow Estimator | DAVIS2016 | SegTrackv2 | FBMS-59 |
|:---:|:---:|:---:|:---:|
| RAFT | 73.9 | 62.2 | 61.3 |
| ARFlow | 59.2 | 51.1 | 50.0 |

Table 9: Ablation studies on flow estimator.

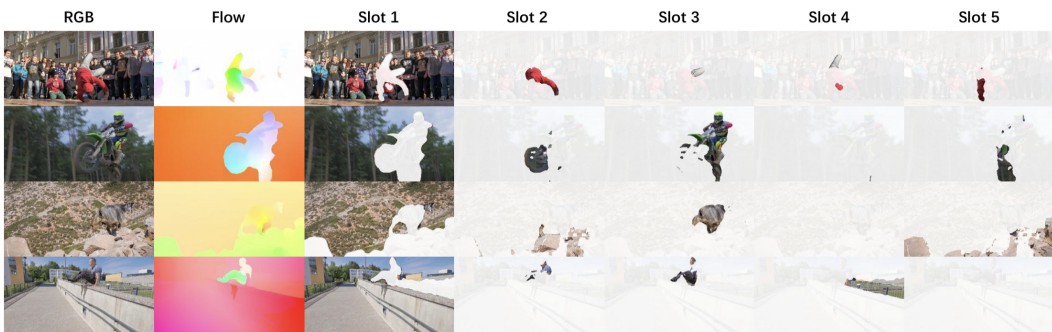

Figure 4: Per-slot visualization on DAVIS 2016.

## C    MORE QUANTITATIVE RESULTS

Besides RAFT flow estimator, we consider a fully unsupervised model ARFlow (Liu et al., 2020) shown in Table 9. The inferior result brought from ARFlow demonstrates that a good optical flow quality lays strong foundation for the success of the object discovery, which aligns with our motivation.

## D    MORE QUALITATIVE RESULTS

We visualize the segmentation results when initializing 5 slots vectors in Figure 4. The results are promising, where each slot groups similar part under the only supervision of motion. We believe more supervision like semantics (Caron et al., 2021) and depth (Ranftl et al., 2021) would lead to better segmentation. We show more qualitative results of SegTrackv2 and FBMS-59 in Figure 5 and Figure 6. Besides,

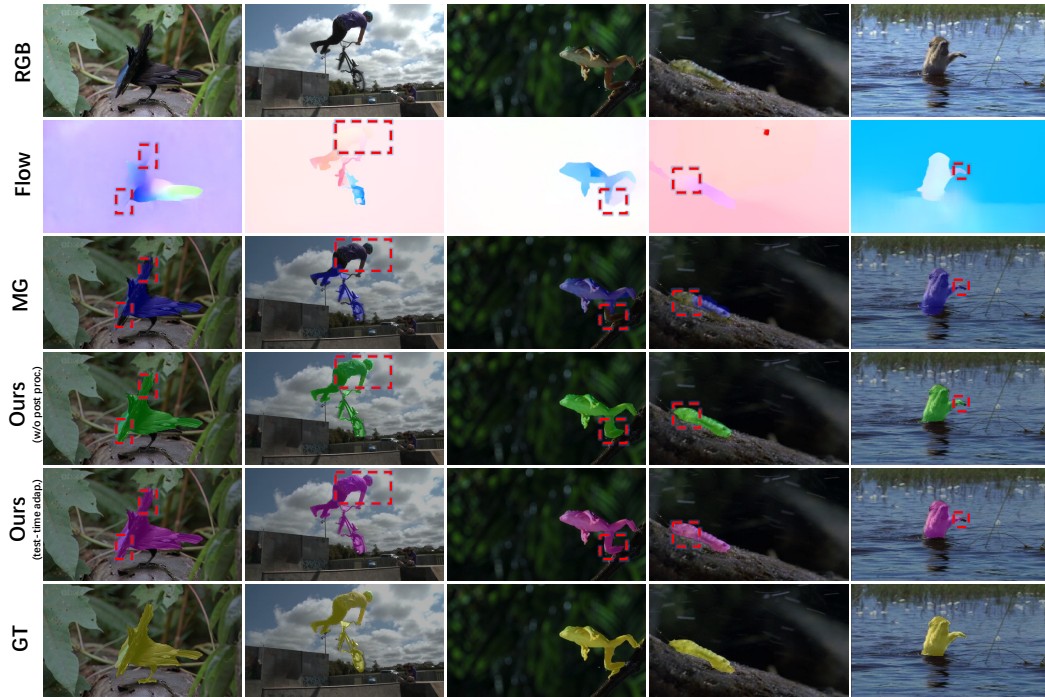

Figure 5: Qualitative results on SegTrackv2. MG refers to Yang et al. (2021a). Red boxes outline the corresponding difference.

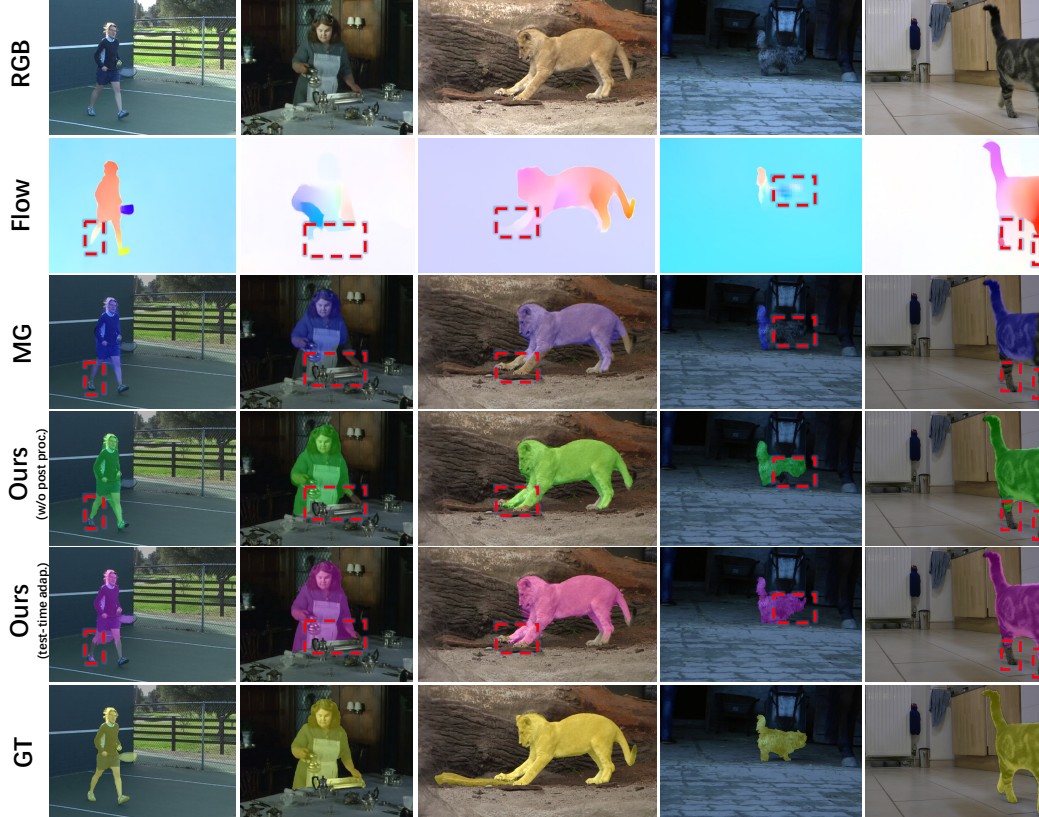

Figure 6: Qualitative results on FBMS-59. MG refers to Yang et al. (2021a). Red boxes highlight the corresponding difference.

