# OpenReview forum: "Motion-inductive Self-supervised Object Discovery in Videos"
_ICLR.cc/2023/Conference — Submitted to ICLR 2023_

### Official Review · Reviewer_HWKn · 2022-10-23

**Confidence:** 4
**Clarity, Quality, Novelty And Reproducibility:** Please see the strength and weaknesse…
**Correctness:** 2
**Technical Novelty And Significance:** 2
**Empirical Novelty And Significance:** 2
**Recommendation:** 3

**Strength And Weaknesses:**

** Strengths
1. Directly relying on RGB frames to overcome the posed limitations from the usage of optical flows
2. Combining several components from other literature to achieve state-of-the-art performances, e.g. iterative routing for layered representation from MG (Yang et al., 2021a) and test-time adaptation from OCLR (Xie et al., 2022)
3. The submitted code and implementation details in the supplementary material facilitate reproducibility

** Weaknesses
1. Mild novelties; The overall framework seems quite similar to the previous idea introduced on MG (Yang et al., 2021a), particularly for dual-layered flow decoder, except for using RGB frames as input and encoding them further through frame comparator. The introduced three loss functions also lack of originalities as they are commonly used in literature for video-based prediction; flow reconstruction as a proxy task for unsupervised learning, temporal smoothness, and contrast of binary masks.
2. Usage of the optical flow; While the authors claim the drawbacks of optical flow in terms of defected supervisory signal, their training object still rely on the pre-computed optical flow, RAFT, that also might suffer from static scenery and occlusions.
3. The need of supervision; I understand that the proposed method do not require supervision for video object segmentation, but the generated optical flow is yielded by the supervised method, RAFT on MPI sintel and KITTI. I would say the required cost to annotate optical flow ground truth for each pixel is much more than the one of binary segmentation. The authors need to demonstrate their method also fairly works well while using unsupervised optical flow methods; such as ARflow.

**Summary Of The Paper:**

The authors proposed a method for unsupervised object discovery in videos. In contrast to the previous methods based on the optical flows between adjacent frames, they directly adopted RGB frame sequences. Specifically, the proposed model first encodes RGB frames into visual features, then pass the paired features into frame comparator module that computes relative motion. Finally, dual-layered flow decoder is introduced inspired by slot attention works that outputs two slots, i.e. foreground and background. Experiments are conducted on three benchmarks, surpassing the previous methods.

**Summary Of The Review:**

As the weaknesses outweigh strengths, I lean towards rejection at this point.

---

> ### Author Response · Authors · 2022-11-16
> **Response to Reviewer HWKn**
>
> Thanks for the suggestion. We would like to claim some points to address the concerns as follows.
>
>
> > Q1: Mild novelties:
>
> A: The motivation is novel compared to MG. Please refer to the general response on novelty.
>
> > Q2: Usage of the optical flow.
>
> A: There is some misunderstanding about the role of optical flow in our paper. We regard the flow as the seed to discover the object out of RGB frames, which we call motion-inductive. We do claim that the single-step optical flow suffers from static scenery and occlusions. Therefore, we take RGB frames as direct input and devise the Temporal fusion module to develop a sense of permanence.
>
> > Q3:  Need of supervision.
>
> A: We have included the results using ARFlow in Table 9. Our method still works fairly on self-supervised flow signals.

---

> ### Author Response · Authors · 2022-11-24
> **Look forward to your reply.**
>
> Dear Reviewer HWKn,
>
> Thanks again for your valuable comments and constructive suggestions, which help us a lot to improve the paper. Please do let us know if our response has addressed your concerns. We are keen to keep engaging with you to discuss any additional questions or comments.
>
> Best wishes,
>
> Authors

---

### Official Review · Reviewer_fGvA · 2022-10-28

**Confidence:** 4
**Correctness:** 3
**Technical Novelty And Significance:** 2
**Empirical Novelty And Significance:** 2
**Recommendation:** 5

**Clarity, Quality, Novelty And Reproducibility:**


Clarity, quality
The paper is clear, the reading is fluid, the approach and its advantage are well discussed. The whole framework is well presented, including graphs and illustrations. The ablation stutThe appendices gives a level of details sufficient enough for the work to be reproduced. The authors will open  the source code.
Visual illustration could be improved by adding the images of the difference between the segmentation output and the object mask ground truth. Since the visual improvements are very local and not ovbious, it would facilitate the interpretation of the visual outputs.
I would personally suggest to put the section Related work after the introduction rather than at the end of the paper. In particular, the work is built upon previous similar and related approach, and it seems to me preferable to mention them at first.

Novelty
The work motivation, ie unsupervised object detection from video without using OF as input, is a continuation of Choudhury and al (2022). From a technical point of view, the network architecture (and loss functions) mainly reuse components which have already proved to be efficient in similar scenarios.


**Details Of Ethics Concerns:**

No  ethic concerns

**Strength And Weaknesses:**


The overall goal of the paper touches to one of the key aspect of modern computer vision, ie unsupervised object detection. Using video as input is indeed a key element, as it enables to separate foreground from background.
One of the interesting observations of the paper (thought not new), is that pairs of frames are not sufficient to accurately detect the motion and hence the object: each frame should be aware of the whole sequence so that the frame-wise segmentation accounts for the whole sequence global motion. This is enabled by the transformer \Phi_temp, which fuses altogether the frames embeddings.

The work attempts to move towards fully unsupervised training, but does in fact use some level of supervision via the OF loss. Similar scenario had been proposed by Choudhury and al (2022). An other choise could have been, instead of supervising the Flow reconstruction loss,  to minimize the image reconstruction loss, after wrapping one of the frames pair to the other. Other proxies to optimize and to constrain the estimated flow could also have been used.

I am intrigued by the behavior of the segmentation masks. I am not sure what prevent the segmentation masks to be uniform in all frames (all ones and all zeros, for s=1,2 or reciprocally). Uniform masks in all frames would lead to an L_entro and L_cons to be zero, which is a situation which could be favored by the network, unless other constraints are given.

In the experimental section, I am not sure what are the results reported for OLCR.  Xie and al 2022 report results without and with test-time adaptation. Only results without adaptation are reported. However, results with test-time adaptation, which the authors also use, outperform the authors results for SegTrack and FBMS, and are very close for DAVIS2016.

**Summary Of The Paper:**


The paper tackles the task of unsupervised moving objects detection from videos. The approach takes as input the video sequence, without flow information.  The network consists of a sequence of known components, ie encoder, transformers, and slot attention decoder, and output a three layer map, ie, the segmentation mask and the bidirectional optical flow. It is only partially un/supervised, given the fact that the main loss function compares the output optical flow to a ground truth one, the lastest one resulting from a fully supervised approach. One of the motivations of the work is to address the problem of 'weak motion', ie when the object might be temporally static in a few frames of the input sequence, or when there are occlusions.  Experimental results are reported on three datasets and compared with related work.

**Summary Of The Review:**

The main strength of the paper is certainly the clarity and quality of the work.  It is in the continuation of Choudhury and al (2022) (both for the scenario and the quantitative results).

---

> ### Author Response · Authors · 2022-11-16
> **Response to Reviewer fGvA**
>
>
> We thank the reviewer for the recognition of the clarity and presentation of our paper. We would like to claim some points to address the concerns as follows.
>
>
> > Q1: Other proxies to optimize the training.
>
> A: We attempt to remove the supervision of OF. In the given framework, pixel-level reconstruction loss after warping makes the training diverge. We analyze it may attribute to the training from scratch. Future work can be explored on the feature-level constraint and more sophisticated training recipe.
>
> > Q2: Uniform masks in all frames could be favored by the network, unless other constraints are given.
>
> A: Well, the situation prevents the constraint by reconstruction loss $L_{\text{recon}}$. That is, the decoder cannot reconstruct the correct flow merely by one of two competing and orthogonal slots.
>
> > Q3: Results for OCLR.
>
> A: For better presentation, we insert one more 'p.p.' column in Table 2 to indicate whether the method uses sorts of post-processing tricks. Yes, the results of OCLR are slightly better than ours for SegTrack and FBMS, but the comparison is not that fair since OCLR trains the proposed model on synthetic training data.
>
>
> > Q4:  Position of related work.
>
> A: Thanks for the advice. We put the section just after the introduction.
>
> > Q5:  Continuation of Choudhury et al (2022).
>
> A: Though both works take RGB images as input and supervision from optical flow, the intuition and methodology are quite different.
> In Choudhury et al (2022), the motivation is to employ optical flow as pseudo-masks for object segmentation. Its input is only one still image (not an image pair) and the output is segmentation results. The model is unable to build temporal coherency for object discovery. Thus, the supervision of optical flow here is utilized as still pseudo-masks to train its image segmentation network (UNet/MaskFormer) via the approximation of the Gestalt principle of common fate. In contrast, our motivation is to establish temporal coherency and object permanence among consecutive frames with the induction of motion. As for the methodology, we encourage the model to find the motion patterns of objects and backgrounds among consecutive frames with the guidance of the optical flow. We obtain segmentation results as a by-product by modeling the dual-layered motion pattern.

---

> ### Author Response · Authors · 2022-11-24
> **Look forward to your reply.**
>
> Dear Reviewer fGvA,
>
> Thanks again for your valuable comments and constructive suggestions, which help us a lot to improve the paper. Please do let us know if our response has addressed your concerns. We are keen to keep engaging with you to discuss any additional questions or comments.
>
> Best wishes,
>
> Authors

---

### Official Review · Reviewer_NSCm · 2022-10-31

**Confidence:** 3
**Correctness:** 3
**Technical Novelty And Significance:** 2
**Empirical Novelty And Significance:** 2
**Recommendation:** 5

**Clarity, Quality, Novelty And Reproducibility:**

Clarity: This paper is well-written and easy to follow.

Quality: Overall, this paper has good quality.

Novelty: This work adds two more modules(i.e., Temporal fusion and frame comparator) on top of Motion Grouping model to get slightly better results, which is not novel enough for me.

Reproducibility: I believe the work is reproductive. Code was provided but  not carefully checked.

**Strength And Weaknesses:**

Strength:
- This paper is well-written and easy to follow.

- The motivation for using RGB as input instead of flow is clear and reasonable, i.e., having cues when static or partially occluded and keeping rich texture information.

- The proposed model architecture and temporal consistency loss have been aligned with these motivations. I like the idea of directly using the RGB frames as input to segment the object as the optical flow is implicitly included in a sequence of the RGB frames. With the design of temporal fusion and frame comparator module, the model can learn optical flow implicitly instead of precomputing it explicitly. The design of temporal consistency loss help to deal with the static or partially occluded situation.

Weaknesses:
- I read your supplementary material about using DINO for test-time adaptation results, but it is unclear to me how do you manage to use the feature from DINO to do mask prop. If you use the idea of mask propagation. Will the mask error be accumulated along the frame? Why it can boost performance a lot?

- Although the proposed method provides several ablation studies, I still suggest the authors conduct the following ablation studies to enhance the quality of the paper:

    * Does the number of learnable queries in the slot attention module impact the performance? Since the slot attention module is initially designed for segmenting different objects in a toy dataset. If there are more queries, are we able to tackle the limitation of segmenting multiple objects separately?

    * Have you tried other optical flow methods such as PWC-Net or ARFlow. It seems that the RAFT optical flow results are very noisy on SegTrackv2 and FBMS-59 datasets using the script provided by motion grouping. Sometimes it is even enabled to identify the object for instance with frame gap =1,2 when the motion changes are too small. Do you think it will hinder your learning?

- A minor question regarding the sliding window during inference. Is that an overlapping sliding window or a non-overlapping sliding window?

- As you mentioned in the limitation, the proposed method is not able to segment multiple objects. Can you put a figure to visualize how it fails (i.e. only the main object is segmented or multiple objects are partially segmented)?

- Some related work is missing:

    * Yanchao Yang et al. Dystab: Unsupervised object segmentation via dynamic-static bootstrapping. In CVPR, 2021.

    * Vickie Ye, et al. Deformable sprites for unsupervised video decomposition. CVPR, 2022.

    * Yangtao Wang, et al. TokenCut: Segmenting Objects in Images and Videos with Self-supervised Transformer and Normalized Cut. arXiv preprint arXiv:2209.00383, 2022


**Summary Of The Paper:**

This paper proposes a self-supervised model for video object discovery. It only takes consecutive RGB frames as input instead of optical flows as in previous works. The model is trained to reconstruct the optical flow between any paired frames that generated through an off-the-shelf optical flow estimator. A temporal consistency loss on motions at different paces is proposed to encourage the model to segment the objects even if they stop moving or are partially occluded at some time point. Experimental results demonstrate its superior performance on several datasets.

**Summary Of The Review:**

This work on self-supervised learning video object discovery using RGB frames only is well-motivated and designed. However, the proposed model seems to add two more modules on top of the Motion Grouping model and the novelty is limited. Although several empirical results have demonstrated the performance on several datasets, more experiments are still needed.

---

> ### Author Response · Authors · 2022-11-16
> **Response to Reviewer NSCm**
>
> We thank the reviewer for the recognition of the motivation and presentation of our paper. We would like to claim some points to address the concerns as follows.
>
> > Q1: How do you manage to use the feature from DINO to do mask prop? Will the mask error be accumulated along the frame? Why it can boost performance a lot?
>
> A: Yes, we use the idea of mask propagation. To deal with the accumulative mask error, we design a key frame selection strategy to filter out a set of keyframes whose object masks are highly confident. In the mask propagation stage, conventional semi-supervised object segmentation only relies on the first frame as a keyframe, while we have multiple keyframes on different temporal positions to correct the accumulated propagation error. In this way, the propagation enhances object permanence across time and boosts performance. We add the clarification in the updated version.
>
> > Q2: Does the number of learnable queries in the slot attention module impact the performance? If there are more queries, are we able to tackle the limitation of segmenting multiple objects separately?
>
> A: Thanks for the suggestion. We conduct a preliminary experiment with 5 learnable queries shown in Figure 4. The results reveal that the separation sometimes vaguely tells the multiple parts of the foreground but not precisely. It is reasonable since due to the nature of flow there are no semantic cues for the model to separate the multiple objects whose motion patterns are the same. The potential integration with ImageNet-pretrained backbone (e.g., DINO) and flow supervision may solve the ambiguity.
>
> > Q3: Have you tried other optical flow methods such as PWC-Net or ARFlow? It seems that the RAFT optical flow results are very noisy on SegTrackv2 and FBMS-59 datasets. Do you think it will hinder your learning?
>
> A: Yes. RAFT estimator might generate noisy flow without clear moving foreground on the SegTrackv2 and FBMS-59 datasets from time to time. And it hinders the success of segmentation. However, we find the generation quality of ARFlow is even worse and leads to inferior results shown in Table 9.
>
> > Q4: A minor question regarding the sliding window during inference. Is that an overlapping sliding window or a non-overlapping sliding window?
>
> A: We utilize overlapping sliding windows, which is clarified in the implementation details now.
>
> > Q5: Can you put a figure to visualize how it fails (i.e. only the main object is segmented or multiple objects are partially segmented)?
>
> A: Yes. We visualize it in Figure 4.
>
> > Q6: Some related work is missing.
>
> A: Thanks. We append those three works in the related work section.

---

> ### Author Response · Authors · 2022-11-24
> **Look forward to your reply.**
>
> Dear Reviewer NSCm,
>
> Thanks again for your valuable comments and constructive suggestions, which help us a lot to improve the paper. Please do let us know if our response has addressed your concerns. We are keen to keep engaging with you to discuss any additional questions or comments.
>
> Best wishes,
>
> Authors

---

### Official Review · Reviewer_7qqQ · 2022-10-31

**Confidence:** 4
**Correctness:** 4
**Technical Novelty And Significance:** 4
**Empirical Novelty And Significance:** 3
**Recommendation:** 8

**Clarity, Quality, Novelty And Reproducibility:**

I would prefer the "related work" section to be put before the modeling section, rather than just before conclusion. But the paper is clearly written and well presented overall. The proposed method is novel in my opinion.

**Strength And Weaknesses:**

Strength:
1. The proposed method has superior performance on public datasets compared to state-of-the-art methods with similar constraint(Liu et al., 2021).
2. The proposed method has close or superior performance on public datasets compared to state-of-the-art methods with less constraints, e.g. using optical flow in the inference stage, use synthetic training data. This also proves the effectiveness of the proposed method.
3. The proposed method effectively exploits temporal fusion so motion cue from multiple frames could be used to discover the object. This makes the method being robust in the cases where object is partially occluded, or is static in partial frames.
4. The proposed method is faster in the inference stage compared to other methods, thanks to the setting of operating on raw RGB video clips.
5. The authors did ablation study to show the impact of each component of the proposed model.

Weaknesses/Questions:
    1. The proposed method is memory intensive. Is this the reason that only up to 7 consecutive frames are used in the experimental setting section?
    2. If the foreground object only appears in part of the video(we don’t have supervisory signal so we don’t have information on this), would the method still work? What would the alpha value for foreground be in those frames where foreground doesn’t exist?
    3. Since optical flow is the training signal, how would the OF quality/failure impact the result? It might be good to evaluate on a synthetic dataset and compare the training with both ground truth OF and estimated/flawed OF.

**Summary Of The Paper:**

This paper proposed a novel method for self-supervised object discovery from video. The authors cast the object discovery problem as optical flow construction problem, where optical flow provided by an off-the-shelf method(RAFT) is used as training signal. The authors demonstrate the effectiveness of the proposed method on several popular datasets includes DAVIS2016, SegTrackv2 and FBMS-59 by comparing to state-of-the-art methods. Compared to other self-supervised object discovery methods exploiting motion cue, the proposed method directly operates on RGB input domain without optical flow computation in the inference stage, thus has advantage on inference speed.

**Summary Of The Review:**

In this paper the proposed object discovery method is practical and novel. The authors did a good job on modeling the problem, perform experiment on standard datasets and doing ablation study. I would recommend acceptance of this paper.

---

> ### Author Response · Authors · 2022-11-16
> **Response to Reviewer 7qqQ**
>
> We thank the reviewer for the recognition of the novelty, presentation, and evaluation of our paper. We would like to claim some points to address the concerns as follows.
>
> > Q1: The proposed method is memory intensive. Is this the reason that only up to 7 consecutive frames are used in the experimental setting section?
>
> A: Yes. In specific, the computation for temporal fusion (i.e., standard transformer block) would be unfriendly and it might be better to adopt pooling-based attention (e.g., MViT) to reduce the memory footprint.
>
> > Q2: If the foreground object only appears in part of the video, would the method still work? What would the alpha value for the foreground be in those frames where foreground doesn’t exist?
>
> A: Whether the pixel belongs to the foreground or background depends on the slot routing and the following flow decoder. The two components theoretically allow that all the screen belongs to the background, where the alpha value for the foreground should be close to zero since we normalize the $\alpha$ via softmax. Empirically, we select some frames with only background and find that the foreground alpha value is minimal as expected. We update the above clarification in section 3.3 of the revised version.
>
> > Q3: Since the optical flow is the training signal, how would the OF quality/failure impact the result?
>
> A: The intuition of the paper is to leverage motion to induct object discovery. Thus, good OF quality lays a strong foundation for the success of the segmentation. To verify it, we conduct the experiment using a fully-unsupervised method ARFlow as a flow estimator and tabulate the result in Table 9 in the supplementary material. Though our method achieves fairly robust results, it is still inferior to the counterpart supervised by RAFT.
>
> > Q4: Position of the "related work" section.
>
> A: Thanks for the suggestion. We have moved the "related work" section forward.

---

> ### Author Response · Authors · 2022-11-24
> **Look forward to your reply.**
>
> Dear Reviewer 7qqQ,
>
> Thanks again for your valuable comments and constructive suggestions, which help us a lot to improve the paper. Please do let us know if our response has addressed your concerns. We are keen to keep engaging with you to discuss any additional questions or comments.
>
> Best wishes,
>
> Authors

---

### Author Response · Authors · 2022-11-16
**General Response on Novelty**

Three reviewers (NSCm, fGvA, HWKn) have questioned the novelty from different perspectives. We understand that the first glimpse of our paper seems similar to MG (Yang et al., 2021a) or GWM (Choudhury et al., 2022). However, the starting point of our work is novel. That is, we aim to build temporal coherency and segment the foreground objects from **consecutive frames** with the induction of motion (optical flow) instead of the single-step flow as input (MG) or single image as input (GWM). Besides, for the module design and loss choice, it is true that we closely follow the setting of MG without considerable modification. The novelty comes from the verification of our intuition that the association of contextual RGB frames exploits the property of **object permanence** while single-step flow fails to do so thus over-relying on the current flow’s object boundary shown in Figures 1 and 3. We supplement the detailed comparison in the related work section of the updated version.

---

### Decision · Program_Chairs · 2023-01-20

**Decision:**

Reject

**Justification For Why Not Higher Score:**

While the paper demonstrates some  promising empirical results, it makes very minor changes (changing the input modality from flow to RGB image pair) in context of prior work.

**Justification For Why Not Lower Score:**

N/A

**Metareview: Summary, Strengths And Weaknesses:**

While this paper demonstrated impressive empirical results for the task of self-supervised object discovery, there were concerns regarding the contributions in context of prior work (MG (Yang et al., 2021a) or GWM (Choudhury et al., 2022)). More concretely, compared to prior work, this paper uses a similar overall approach, with the distinction that the input modality varies from optical flow to pairs of RGB images. While the author response clarifying this distinctions with prior works is noted and the empirical benefits of RGB images as input are interesting, the AC agrees with the majority of the reviewers that these are not major contributions.